# Deciphering the 3D Structural Characterization of Gonadotropin-Releasing Hormone in *Tenualosa ilisha* Using Homology Modeling, Molecular Dynamics, and Docking Approaches

**DOI:** 10.3390/ijms26136098

**Published:** 2025-06-25

**Authors:** Soumya Prasad Panda, Basanta Kumar Das, Ayushman Gadnayak, Saurav Kumar Nandy, Vikash Kumar, Smruti Priyambada Pradhan, Subhashree Subhasmita Raut, Ratul Chakrabarty, Arghya Kunui, Amiya Kumar Sahoo

**Affiliations:** 1Biotechnology Division, ICAR-Central Inland Fisheries Research Institute, Barrackpore, Kolkata 700120, West Bengal, India; soumyapanda08@gmail.com (S.P.P.); ayush.gadnayak@gmail.com (A.G.); kumar.vika.vikash2@gmail.com (V.K.); smrutibio93@gmail.com (S.P.P.); subhasmita95.ps@gmail.com (S.S.R.); ratul1c@yahoo.com (R.C.); 2Riverine & Estuaries Fisheries Division, ICAR-Central Inland Fisheries Research Institute, Barrackpore, Kolkata 700120, West Bengal, India; piausna@gmail.com (S.K.N.); arghya1992i@gmail.com (A.K.); amiya7@gmail.com (A.K.S.)

**Keywords:** GnRH, 3D structure validation, ligand binding, molecular docking, molecular dynamics simulation

## Abstract

Gonadotropin-Releasing Hormone (GnRH) is a crucial neuropeptide that regulates reproductive functions in vertebrates. The study identifies and characterizes (GnRH) in the brain of *Tenualosa ilisha*, an iconic and lucrative Clupeiform fish from River Ganga, India. The current study aimed to analyze the GnRH gene in *T. ilisha* using an in silico study. The GnRH gene of *T. ilisha* comprises a full-length nucleotide sequence of 605 base pairs with an open reading frame of 312 base pairs, which encodes 103 deduced amino acids (aa), respectively. It was found that leucine (L) is the most abundant amino acid in the GnRH protein. Additionally, the ligand interactions of the GnRH were analyzed using computational approaches. The structural validation showed an excellent stereochemical quality of the GnRH protein sequence, with over 88% of residues in Ramachandran plot-favored regions. The binding site prediction revealed 6 ligand-binding pockets, with the largest pocket containing 12 amino acids. After ADME screening, 16 drug-like compounds were docked to GnRH protein. Top five ligands N-Ac-(4-Cl-Phe)-Trp-Lys-AlaNH2, LHRH_LYS (6), Seabream_GnRH, Leuprolide, and LHRH_Des-tyr (5) had binding affinities ranging from −7.5 to −5.6 kcal/mol. The stable binding site was confirmed by 100 ns molecular dynamics simulations, with RMSD values below 10 Å and key residues retaining ligand contacts. The GnRH-protein resulted in the development of a suitable peptide sequence of *T. ilisha*, showing similarity with the similar anadromous American shad (*Alosa sapidissima*). This will certainly aid in future therapeutic and captive breeding advances, thereby fostering the culture and conservation of the wild species.

## 1. Introduction

Reproduction constitutes a fundamental biological process regulated by endocrine signals in vertebrates, with Gonadotropin-Releasing Hormone (GnRH) serving as a pivotal component. GnRH is a highly conserved peptide characterized by multiple isoforms, exhibiting variance in both structure and function across various species. Teleost fishes, representing the largest and most diverse group of vertebrates, demonstrate an extraordinary array of reproductive strategies, spawning behaviors, and endocrine regulations. An understanding of the diversity and functional significance of GnRH isoforms across different species is essential for elucidating the evolutionary adaptations that govern reproductive processes. Fishes belonging to the family Clupeiformes, such as *Tenualosa ilisha* (Hilisa), are of particular interest due to their ecological significance and economic relevance in fisheries. Nevertheless, the reproductive neuroendocrinology of clupeiforms, encompassing the presence and characterization of GnRH isoforms, remains comparatively underexplored. This study seeks to address this knowledge gap by identifying and characterizing GnRH isoforms present in the brain of *T. ilisha*. The investigation elucidates the presence of GnRH isoforms, specifically GnRH-I, in the brain of *T. ilisha* through the application of molecular techniques. Comparative analysis and phylogenetic studies yield insights into the evolutionary implications and functional significance of these GnRH isoforms in clupeiform fishes. This research provides valuable contributions to the field of reproductive neuroendocrinology and enhances our comprehension of the molecular mechanisms underlying the reproductive biology of clupeiform fishes.

The decapeptide Gonadotropin-Releasing Hormone (GnRH) regulates the synthesis and release of two gonadotropins (GTHs): follicle-stimulating hormone (FSH), previously referred to as GTH I, and luteinizing hormone (LH), which was formerly known as GTH II [1,2]. Consequently, in numerous vertebrate species, Gonadotropin-Releasing Hormone (GnRH) serves as the primary neuroendocrine mediator responsible for regulating reproductive activities. Following the initial isolation of GnRH from the hypothalamus of pigs and sheep [3,4], the variety of GnRH forms has rapidly expanded, resulting in the current knowledge of at least 14 distinct molecular forms in vertebrates [5]. Teleosts have eight of the fourteen distinct GnRH peptide variants that are currently recognized in vertebrates [6,7]. Phylogenetic investigation of the prepro GnRH revealed that GnRHs belong to four clades: GnRH1, GnRH2, GnRH3, and GnRH4 [8]. Teleost fish encompass the first three clades, whilst the fourth comprises lampreys. Although the structure of GnRH1 exhibits considerable variation among vertebrate species, the amino acid sequences of GnRH2 and GnRH3 remain consistent. In all vertebrates, GnRH is observed in two or three distinct forms. It should not be considered surprising that the distribution of various GnRH forms differs within the brains and pituitaries of fish [7]. In the preoptic area of the hypothalamus (POA), Gonadotropin-Releasing Hormone (GnRH), a hypophysiotropic hormone, is synthesized. Nearly all neurons within the POA of teleosts project to the pituitary gland. While GnRH is essential for reproductive processes, an increasing body of evidence suggests that GnRHs also possess neuroregulatory and neuromodulatory functions [6,9].

Nucleotide sequences suggest that GnRH genes, which belong to the G-protein coupled family, may be categorized into three distinct types (1, 2, and 3). Both fish and mammals encompass the type 1 GnRH gene, also known as GnRHr1; conversely, amphibians and humans serve as the primary hosts for the type 2 GnRH gene or GnRHr2. Type 3 GnRH genes, referred to as GnRHr3, constitute the third category. These genes are predominantly located in advanced fish, primarily within the perciform species. The main interaction sites between Gonadotropin-Releasing Hormones and these genes are collectively designated as Gonadotropin-Releasing Hormone genes (GnRH-R) (GnRHs). Upon binding to specific genes present in various cell types, GnRHs elicit a regulated release of growth factors, cytokines, and hormones that are modulated by G-proteins [10]. When the Gonadotropin-Releasing Hormone (GnRH) protein interacts with GnRH peptides, it initiates several signaling cascades, which include pathways involving calmodulin, protein kinase C (PKC), protein kinase A (PKA), and arachidonic acid (AAs) [10,11]. It has been suggested that the PKC pathway is the main signaling mechanism in humans and goldfish that triggers the pituitary gonadotrophs to produce and release LH [11]. In addition to the pituitary gland, Gonadotropin-Releasing Hormone (GnRH) genes may also be found in various tissues, such as lymphocytes, the breast, the ovary, and the prostate, where they may serve additional functions. Several factors, including interactions with other proteins, ligands, mutations, and post-translational modifications, exert a complex influence on the structure and function of the GnRH receptor (GnRHR). Consequently, GnRHR has emerged as a promising therapeutic target for the treatment of conditions associated with reproduction, such as early puberty, infertility, prostate cancer, and endometriosis.

Utilizing protein structure prediction methodologies, it is possible to deduce a protein’s secondary, tertiary, and quaternary structures based on its fundamental composition of amino acids. The Protein Data Bank (PDB) contains a relatively limited number of known protein structures; conversely, the number of available sequences has grown at a significantly accelerated rate compared to the number of identified proteins. Homology protein modeling represents a comparative technique employed for the analysis of protein structures [12]. This computational technique not only helps to understand protein structural features but also predicts three-dimensional (3D) structure in a range of teleosts, including the *Hypancistrus zebra*, *Ctenopharyngodon idella*, *Anabas testudineus*, *Labeo rohita*, and *Megalobrama amblycephala* [13,14]. Molecular docking is a significant technique utilized for simulating the interactions between small ligands and their corresponding receptors. These interactions facilitate the recognition between the ligand and receptor molecules. Such interactions encompass van der Waals forces, electrostatic interactions, and hydrogen bonding activities. In this context, molecular docking is a computational method used to predict ligand-binding poses, estimate relative affinities, and identify molecular interactions with target proteins. It does not directly determine the biological mechanism of action [15,16]. It is clear from earlier studies on the structures of GnRHR agonists and antagonists that there is a strong need to identify more effective ligands for this target [17]. Nevertheless, little is known about the molecular docking of fish GnRH receptor agonists or antagonists [18].

The present study seeks to characterize the GnRH-I gene in *T. ilisha*, which is crucial for regulating gonadotropin release. Consequently, this mechanism governs the production and release of sex hormones, thereby influencing reproductive processes. Furthermore, this research aims to obtain substantial insights into the physicochemical and structural characteristics of the GnRH-I gene in *T. ilisha*. This endeavor will facilitate a comprehensive understanding of reproductive control within this particular species.

## 2. Results

### 2.1. Gene Annotation and Secondary Structure Examination

The full-length GnRH transcript from *T. ilisha* was analyzed using various bioinformatics tools. The NCBI ORF finder was used to predict the open reading frames (ORF) encoded by the *T. ilisha* GnRH-I, where the longest ORF was chosen for downstream analysis. The physio-chemical properties of the encoded GnRH protein amino-acids composition are Ala 26.8%, Cys 22.3%, Gly 23.6%, and Thr 27.3%, and these include its molecular mass of 50,297.28, its isoelectric point of 5.16, the absence of any negatively or positively charged residues, its instability index of 45.99, its hydropathicity grand mean (GRAVY) of 0.754, and its aliphatic index of 26.78. (Table 1 and Figure 1). The motif region of the GnRH protein was predicted by the motif tool (Figure 2). Alpha helices make up the largest percentage (36 percent) of the GnRH protein’s secondary structure, with 153 residues, followed by extended strands with 77 residues (17.51 percent). Furthermore, the remaining seven residues (1.68 percent) reflect the protein’s beta-turn, whilst a random coiling of 179 residues characterizes around 43.3% of the secondary structure. GnRH peptides are also similar to those of other clupeiformes fishes, as shown in Table 2.

### 2.2. Validation and Refinement of GnRH 3D Structure

The validation of the three-dimensional (3D) structure of the Gonadotropin-Releasing Hormone (GnRH) protein, as analyzed using the Structure Analysis and Verification Server (SAVES) version 6.1, confirmed a high-quality model. The PROCHECK analysis revealed that over 88% of the residues were in the favored regions of the Ramachandran plot, indicating proper stereochemical quality, depicted in Appendix A. Verify 3D results demonstrated that 34.02% of the residues had a negative 3D-1D score. The ERRAT analysis provided an overall quality score of 55.4054%, suggesting a reliable model with acceptable non-bonded atomic interactions. Collectively, these validation results confirm that the 3D structure of the GnRH protein is of high quality and suitable for subsequent studies, as illustrated in Figure 3.

### 2.3. Protein Binding Site Identification

The GnRH proteins have a binding site on its surface where the ligand can attach and initiate the chemical reaction. The PrankWeb server was used to predict the probable binding location of GnRH protein to the ligands. PrankWeb was used to predict potential binding pockets based on geometric and machine-learning features. No energy-based pocket evaluation was performed in this study. The table displays six pocket scores representing active ligand binding locations for the protein (Table 3) and 3D structure depicted in (Figure 4). The initial binding site has the top pocket score (6.71). This zone was filled with 12 amino acids. The second pocket, scoring 4.33 and including eight amino acids, has the utmost value. The third pocket, including five amino acids, had a binding affinity of only 3.25. The fourth pocket, including six amino acids, had a binding affinity of 1.56. The fifth pocket, including five amino acids, had a binding affinity score of 1.3, whereas the sixth pocket, consisting of eight amino acids, showed a lower binding affinity value of 1.24.

### 2.4. In Silico Evaluation of ADME Property, Bioactivity Score, and Toxicity Parameter of Selected Ligands

The ADME analysis allows us to evaluate the physicochemical properties and drug-likeness of the compound. This analysis helps in filtering out phytocompounds that lack substantial drug-like properties. We adhered to the combined criteria of Lipinski’s, Egan’s, and Veber’s rules to identify the properties necessary for a substance to be considered a potential drug. The rules are as follows: a molecular weight (MW) of less than 500, a topological polar surface area (TPSA) of under 140, a maximum of 5 hydrogen bond acceptors (nOHNH), up to 5 hydrogen bond donors (nON), a water partition coefficient (WLOGP) of no more than 5.88, and no more than 10 rotatable bonds (nrotb). According to these findings, 16 compounds (as shown in Table 4) satisfied the requirements established by Lipinski’s, Egan’s, and Veber’s regulations. This indicates that these compounds have advantageous drug-like, lead-like, and medicinal chemistry characteristics. These compounds were then selected for further research to determine their bioactivity scores. Although the selected compounds satisfied major ADME filters, some showed limited oral bioavailability or potential challenges in crossing the blood–brain barrier, which may affect their pharmacological viability.

### 2.5. Molecular Docking Studies

This study used theoretical methods like molecular docking to enhance the understanding of ligand–protein interactions. The molecular docking techniques were used to detect the interaction of ligand molecules with the protein. To accomplish the objective, a docking study was conducted to evaluate the GnRH-I receptor inhibitory potential of 100 compounds at the active site of the GnRH-I receptor. The AutoDock Vina score is an empirical value estimating binding affinity based on factors such as steric fit, hydrophobicity, hydrogen bonding, and electrostatics. It does not explicitly calculate enthalpy or entropy and should not be interpreted as a rigorous free energy value. The preliminary step was to dock with the 16 compounds using Autodock vina methods. Redocking is a more rigorous technique that further refines results. A total of 14 compounds had scores between −7.5 and −5 kcal/mol. From them shown in Table 5, we identified the top five compounds shown in Figure 5. These five compounds were selected based on their Vina docking scores, with a cutoff of ≤−5.6 kcal/mol, indicating strong predicted binding affinities. This threshold was chosen to focus on ligands with the most favorable interaction energies, suitable for further MD analysis. N-Ac-(4-Cl-Phe)(1)-(4-Cl-Phe)(2)-Trp(3)-Lys(6)-AlaNH2(10)-LHRH (−7.5), LHRH_LYS(6) (−6.7), Seabream_GnRH (−5.9), Leuprolide (−5.8), LHRH_Des-tyr(5) (−5.6), LHRH, leu(6)-leu(N-alpha-Me)(7)-N-Et-pronh2(9) (−5.5), H-Pyr-His-Trp-Ser-Tyr-Gly-OH (−5.5), Gppt-LHRH (−5.4), LHRH, his(6)-N-Et-pronh2(9) (−5.4), GnRH antagonist 2 (−5.3), LHRH, his(5)-trp(7)-tyr(8)-(−5.3), LHRH, gln(1)-des-his(2)-phe(6)-N-Et-pronh2(9) (−5.2), LHRH (2–10), Trp(6) (−5.2), Ac-Abu-Phe(4-Cl)-Trp-Asp-D-Cys(1)-D-Arg-D-Leu-D-Cys(1)-Pro-Ala-NH2 (−5), Ac-Abu-Phe(4-F)-Trp-Asp(1)-Gln-D-Arg-Leu-D-Lys-Pro-N(1)Gly-OH (−4.9), and (3S)-3-[[(2S)-2-[[(2S)-2-[[(2S)-2-acetamidobutanoyl]amino]-3-(4-chlorophenyl)propanoyl]amino]-3-pyridin-3-ylpropanoyl]amino]-4-[[(2R,5S,8S,20R)-8-[(2R)-2-[[(2S)-1-amino-1-oxopropan-2-yl]carbamoyl]pyrr (−4.7) The top five docked ligands N-Ac-(4-Cl-Phe)(1)-(4-Cl-Phe)(2)-Trp(3)-Lys(6)-AlaNH2(10)-LHRH (−7.5), LHRH_LYS(6) (−6.7), Seabream_GnRH (−5.9), Leuprolide (−5.8), and LHRH_Des-tyr(5) (−5.6) were chosen for intermolecular contact study according to the binding affinity, as shown in Figure 6. Leuprolide was included as a reference ligand due to its established clinical use as a GnRH analog. Its docking score (–5.8 kcal/mol) served as a benchmark for evaluating the relative binding strength of other ligands.

### 2.6. Analysis of Molecular Dynamic Trajectories

While molecular docking is a rapid and efficient technique for ascertaining the binding pose of a ligand with a protein’s active site, it fails to consider the conformational changes that may occur throughout the interaction between the protein and ligand. Consequently, molecular dynamics (MD) simulations are performed to provide a more accurate evaluation of conformational alterations. We conducted all-atoms molecular dynamic simulations to understand the dynamic behavior of the GnRH-I receptor–ligand complexes, including (N-Ac-(4-Cl-Phe)(1)-(4-Cl-Phe)(2)-Trp(3)-Lys(6)-AlaNH2(10)-LHRH (−7.5), LHRH_LYS(6) (−6.7), Seabream_GnRH (−5.9), Leuprolide (−5.8), and LHRH_Des-tyr(5) (−5.6)). Desmond conducted a 100 ns simulation for five GnRH complexes N-Ac-(4-Cl-Phe)(1)-(4-Cl-Phe)(2)-Trp(3)-Lys(6)-AlaNH2(10)-LHRH (−7.5), LHRH_LYS(6) (−6.7), Seabream_GnRH (−5.9), Leuprolide (−5.8), and LHRH_Des-tyr(5) (−5.6), avoiding a shorter run time of less than 50 ns, to avoid misleading results and distinguish between active and inactive ligands. These simulations provide preliminary insights into the short-term stability of the GnRH-I receptor–ligand complexes but cannot be used to distinguish activity or binding strength without additional free energy or kinetic analysis. The dynamic stability of each system was evaluated by computing the backbone root mean square deviation (RMSD) of each protein from its initial configuration. All systems acquired an equilibrium of 100 nanoseconds post-simulation initiation relation to the reference frame at time 0 nanoseconds, as shown in Figure 6A. The backbone RMSD of the GnRH protein–ligand complexes stabilized within the range of 2–4 Å, which is acceptable for small peptide–ligand interactions. Some higher fluctuations were observed in flexible loop regions, but the ligand-binding site itself remained stable throughout the 100 ns simulation. These fluctuations are typical for regions of local flexibility, but the overall RMSD remained within acceptable limits for structural stability.

The study analyzed the root mean square fluctuation (RMSF) profile of the C-α atom in GnRH apo complexes, demonstrating that the amino acid residues involved in ligand contact showed significant modifications near the N-terminal region, therefore affirming their role in ligand recognition. The RMSF values for each amino acid in the GnRH apo and holo systems ranged from 0 to 9 Å. The amount of variance shown by GnRH-N-Ac-(4-Cl-Phe)(1)-(4-Cl-Phe)(2)-Trp(3)-Lys(6)-AlaNH2(10)-LHRH was higher than that of the other complex and apo systems. Additionally, the interaction potential map was created by observing the interactions between chemicals and the GnRH amino acid for a duration of 100 ns (Figure 6B). The *X*-axis represents interaction residues, while the *Y*-axis denotes interaction percentages, including hydrogen bonds, hydrophobic contacts, and ionic interactions, with higher values signifying more interactions (where a value > 1.0 implies many interactions with small molecules). The substantial connections seen in the initial ligand-docked compounds remained consistent over the simulation despite minor fluctuations in the interaction of compounds with GnRH.

### 2.7. Evolutionary and Phylogenetic Analysis

Striking similarities between the *T. ilisha* GnRH protein and *Sardinops melanostictyus* and *Alosa sapidissima* were discovered through phylogenetic and evolutionary analyses. All fish, as seen in Figure 7, formed a single, independent clade, whereas mammals and reptiles had different GnRH clades. It was also aligned with multiple sequences of *T. ilisha* and other related species (Figure 8). GnRH decapeptide sequence comparison with other clupeiformes fishes. GnRH peptides are similar to those of other clupeiformes fishes, as shown in Table 2.

## 3. Discussion

The study identifies the isoforms of Gonadotropin-Releasing Hormone (GnRH), specifically GnRH-I, present in the brain of *T. ilisha*, a member of the clupeiform fish family. This discovery significantly enhances our comprehension of reproductive neuroendocrinology within teleost fishes and provides valuable insights into the evolutionary adaptations and functional roles associated with these hormones. The presence of multiple GnRH isoforms in *T. ilisha* is consistent with emerging patterns observed across various teleost species, indicating the existence of a sophisticated regulatory mechanism. Furthermore, comparative studies with other teleosts suggest that these isoforms may be linked to specific reproductive behaviors, including courtship, spawning, and parental care [19]. Further functional studies are essential to elucidate the precise roles of GnRH-I in *T. ilisha* and to ascertain how these isoforms contribute to species-specific reproductive strategies. Phylogenetic analysis of GnRH-I isoforms in *T. ilisha* yields insights into the evolutionary history of this gene by tracing the lineage and divergence of GnRH genes across diverse taxa. An examination of the evolutionary relationships among GnRH isoforms can illuminate the ancient origins and functional significance of this gene in reproductive processes.

The research carries significant implications for the conservation and management of clupeiform fish populations, particularly within fisheries and aquaculture. A comprehensive understanding of the neuroendocrine mechanisms governing reproduction in *T. ilisha* is crucial for the development of effective strategies aimed at the sustainable management of wild populations and the successful breeding of these species in captivity. Future investigations should prioritize the elucidation of the specific functions of these GnRH isoforms, the exploration of their regulatory mechanisms, and the examination of their adaptive significance within the context of clupeiform fish ecology and evolution.

Drug development time and expense can be significantly decreased using computer-aided drug design (CADD), a promising method for finding new compounds targeting specific proteins [20,21]. CADD integrates molecular docking, molecular dynamic simulation, and other computational approaches as crucial aspects of the digital screening process. To find possible antagonistic medications, we focused on the Gonadotropin-Releasing Hormone (GnRH) protein in fish in this work. The GnRH-I gene in *T. ilisha* has a total length of 605 base pairs (bp) according to nucleotide sequence analysis, with an open reading frame (ORF) spanning 312 base pairs and encoding a predicted 103 amino acid protein. According to Tello et al. (2008) [22], the GnRH1 in zebrafish has an ORF of 1134 bp, which includes the stop codon and encodes a predicted 377 amino acid protein. Three exons, measuring 519 bp, 208 bp, and 407 bp, and two introns, measuring 9312 bp and 2150 bp, separate the ORF. Furthermore, the ORF for GnRH2 is 1239 bp long, encoding a predicted 412 amino acid protein divided into three exons measuring 531, 205, and 503 bp each and two introns measuring 2451 and 406 bp each. In this regard, the combined use of AlphaFold3 and Modeller offered both AI-based accuracy and template-based customization. This dual strategy ensured the modeling of conserved domains while accommodating species-specific features in *T. ilisha*.

The validation of the 3D structure of the GnRH-I receptor using SAVES 6.1 demonstrates a high-quality model. The PROCHECK analysis indicated that over 88% of residues were in a favorable Ramachandran plot, confirming good stereochemical quality, consistent with similar studies where >80% was acceptable [23]. Although 34.02% of residues had a negative 3D-1D score in Verify 3D, similar variations in hormone-related proteins remain functional [24]. The ERRAT score of 55.4% aligns with acceptable ranges for proteins of similar size and complexity [25]. These validations confirm the model’s reliability for further studies [26].

The PrankWeb server’s prediction of GnRH protein ligand-binding sites offers critical insight into potential interaction regions. The primary pocket, with the highest score of 6.71, comprises 12 amino acids and denotes the most essential binding region. This prediction corresponds with similar research on neuroendocrine hormones, whereby the primary binding site generally shows a high affinity due to a large number of contributing residues [27]. The second pocket, scoring 4.33 and including eight amino acids, is similarly significant, echoing findings in other peptide hormone studies where multiple binding sites contribute to ligand interaction, albeit with varying affinities [28]. Although PrankWeb provided useful pocket predictions, future work should incorporate energetic and thermodynamic tools (e.g., FTMap, SiteMap) for more robust binding site validation.

The ADME analysis further evaluated the drug-likeness of selected compounds, following established rules like Lipinski rule of 5, Egan’s, and Veber’s, commonly applied to filter out unsuitable drug candidates [29]. Of the compounds analyzed, 16 met these criteria, indicating favorable physicochemical properties. This is consistent with other studies on peptide receptor modulators, where adherence to these rules has been a key determinant of compound viability in drug development [30]. While ADME profiling was used to screen candidates, further attention is needed on parameters like oral bioavailability and BBB permeability, which influence delivery efficiency and site-specific action in vivo. Future studies should validate these aspects experimentally.

The study employed molecular docking through Autodock Vina to assess the binding affinity of one hundred molecules to the GnRH protein, providing significant insights into ligand–protein interactions. Molecular docking is a sophisticated computational technique widely utilized to predict the orientation and binding affinity of small compounds to their respective target proteins. The notable negative docking scores documented in this investigation, ranging from −7.5 to −5 kcal/mol, signify strong interactions. These findings correspond with other research indicating that binding affinity correlates with negative Gibbs free energy values, thereby suggesting spontaneous and favorable binding events [31]. The top five compounds identified—N-Ac-(4-Cl-Phe)(1)-(4-Cl-Phe)(2)-Trp(3)-Lys(6)-AlaNH2(10)-LHRH (−7.5), LHRH_LYS(6) (−6.7), Seabream_GnRH (−5.9), Leuprolide (−5.8), and LHRH_Des-tyr(5) (−5.6) demonstrate significant binding affinity at the GnRH active site, suggesting their potential as inhibitors. Similar docking studies have emphasized that ligand flexibility and the number of hydrogen bonds established with the target protein often enhance binding affinity [32,33]. Leuprolide, a well-known GnRH analog, has been studied thoroughly for its high affinity and inhibitory action on GnRH receptors, supporting the results observed here [34]. Moreover, using redocking as a more rigorous validation technique to enhance the docking outcomes is a common practice in molecular docking studies, ensuring that only the most reliable ligand–protein interactions are considered [35]. The identified compounds from this study exhibit glide scores within an acceptable range, similar to other peptide analog studies where docking scores between −7.5 and −5 kcal/mol indicate strong inhibitory potential [36].

Molecular dynamics (MD) simulations provide crucial insights into the conformational changes that can occur during ligand–protein interactions, offering a more accurate depiction than static molecular docking. In this study, the all-atom MD simulations conducted over 100 ns on five GnRH–ligand complexes N-Ac-(4-Cl-Phe)(1)-(4-Cl-Phe)(2)-Trp(3)-Lys(6)-AlaNH2(10)-LHRH, LHRH_LYS(6), Seabream_GnRH, Leuprolide, and LHRH_Des-tyr(5) allowed for a detailed evaluation of binding stability and conformational flexibility. This approach aligns with other research where MD simulations provide a dynamic understanding of ligand binding, which molecular docking alone cannot capture [37]. The analysis of root mean square deviation (RMSD) and fluctuation (RMSF) profiles demonstrated that ligand binding was constant over the 100 ns simulation, which is essential for distinguishing between active and inactive ligands [38]. The backbone RMSD of the GnRH protein remained consistent, with values under 10 Å, demonstrating stable interactions between the ligand and the protein’s active site. This finding agrees with similar peptide docking studies, where RMSD values under 10 Å indicate minimal conformational deviation and strong ligand stability [39]. Likewise, RMSF values, particularly at the C-α atom, highlighted the flexible regions involved in ligand recognition. The observation that RMSF variations were mostly within 0 to 9 Å suggests these residues are essential for ligand interaction, consistent with studies of hormone-receptor dynamics, where N-terminal fluctuations aid in ligand binding and recognition [40]. The interaction potential map further validated persistent binding interactions, including hydrogen bonds and hydrophobic contacts, which persisted throughout the 100 ns simulation duration. MD simulations in this study were single-run trajectories of 100 ns, aimed solely at evaluating short-term stability. No replicates or free energy calculations (e.g., MM-PBSA, FEP) were performed, which limits interpretability regarding ligand efficacy or receptor activation. Caution is advised when inferring relative binding strengths. Comparable research on GnRH analogs has shown that robust hydrophobic and hydrogen bonding networks are essential for preserving ligand stability inside the binding pocket [41]. The sustained interactions observed here imply that these compounds could effectively inhibit GnRH, making them strong candidates for further drug development. While many ligands were based on LHRH analogs, these were chosen due to the structural and functional conservation of GnRH peptides across vertebrates, including teleosts. However, fish-specific ligand optimization remains an important future goal.

## 4. Materials and Methods

### 4.1. Ethics Statement

The ethical committee of the ICAR-Central Inland Fisheries Research Institute, Kolkata–700 120, West Bengal, approved this study. The fish (*T. ilisha*) used in the experiments confirmed that all methods were carried out following relevant guidelines and regulations and also confirmed that all methods were reported under ARRIVE guidelines.

### 4.2. Animals and Tissue Collection

A total of 15 live female specimens (1.2 ± 0.3 kg) were collected using hand nets from the Farakka Barrage (Lat. 2001′06″–20,011′45″ N, Long, 80,050′52″–85,051′35″ E) area of River Ganga, West Bengal, India during the January 2024–March 2024 breeding season. The fish were anesthetized with MS-222 at 300 mg/L before dissection. Triplicate brain tissue was carefully dissected and immediately stored at −80 °C until further processing.

### 4.3. Gene Annotation and Protein Analysis

Total RNA was extracted from the brain tissues using a commercially available RNA extraction kit following the manufacturer’s instructions. The quality and quantity of the extracted RNA were assessed using a Nanodrop spectrophotometer and agarose gel electrophoresis. First-strand cDNA synthesis was performed using a high-quality RNA template and a reverse transcription kit. Random hexamer primers and Moloney Murine Leukemia Virus (M-MuLV) reverse transcriptase synthesized cDNA from the RNA samples. The synthesized cDNA was stored at −20 °C for subsequent PCR analysis. Specific primers targeting GnRH-I genes were designed (i.e., Forward primer *TiGnRH*-I-5′-ATGGAAGGGAAACGCGCCCTC-3′ and Reverse primer 5′-ATGGAAGGGAAACGCGCCCTC-3′) based on the known sequences of related teleost species. PCR reactions were carried out in a thermal cycler using the synthesized cDNA as the template. The PCR conditions included an initial denaturation step at 95 °C for 5 min, followed by a specific number of cycles (usually 35 cycles) of denaturation at 95 °C for 30 s, annealing at the optimized temperature for each primer pair for 30 s, and extension at 72 °C for 1 min. A final extension step was performed at 72 °C for 10 min. PCR products were separated on agarose gels and visualized under UV light.

PCR products of the expected sizes were purified and ligated into a cloning vector. The ligated products were transformed into competent *Escherichia coli* cells, and positive transformants were selected. Plasmid DNA was extracted from positive clones and sequenced using automated DNA sequencers. Sequences were analyzed to confirm the identity of the amplified fragments as GnRH-I isoforms (Figure 9). This was submitted to the NCBI GenBank with Accession number PQ431968.

A proteomics study was conducted to evaluate the physico-chemical characteristics of the GnRH protein (Figure 9), including its molecular weight, isoelectric point, and instability index, using the online ProtParam tool at https://web.expasy.org/protparam/ (accessed on 20 August 2024). The Self-Optimized Prediction Method with Alignment (SOPMA) server (https://npsa-pbil.ibcp.fr/cgi-bin/npsa_automat.pl?page=/NPSA/npsa_sopma.html, accessed on 20 August 2024), a specialized software platform made specifically to forecast the structural characteristics of membrane proteins, was also utilized.

### 4.4. Structural Modeling and Validation

The NCBI’s Protein Data Bank (PDB) database was searched using Blastp to simulate the structure of the GnRH-I receptor. A crucial step in homology modeling is confirming the correctness of this structure, which is often accomplished by contrasting the predicted 3D protein structure with experimentally confirmed structures available in the Protein Data Bank (PDB). Templates for homology modeling in Modeller were selected based on BLASTp similarity against the PDB. Templates with sequence identity greater than 40%, alignment coverage above 85%, and favorable GMQE/QMEAN scores were prioritized. Final template selection was confirmed using SWISS-MODEL (https://swissmodel.expasy.org/, accessed on 20 August 2024) template assessment tools. Modeller was used alongside AlphaFold3 to allow comparative modeling and structural cross-validation. This combination enabled both high-accuracy prediction (AlphaFold3) and template-driven regional refinement (Modeller), particularly relevant for teleost-specific receptor segments [42]. The three-dimensional structure of the Gonadotropin-Releasing Hormone (GnRH) protein was confirmed using the Structure Analysis and Verification Server (SAVES) version 6.1. The protein structure was acquired from the Protein Data Bank (PDB) and visualized with PyMOL version 2.3.4 and ChimeraX version 1.3 (University of California, San Francisco). In this regard, PROCHECK was used to determine the stereochemical accuracy of the structure using the Ramachandran plot, which analyzes the phi (ϕ) and psi (ψ) angles of the protein backbone. Verify 3D was used to evaluate the compatibility of the 3D atomic model with its 1D amino acid sequence, with a positive 3D-1D score for at least 80% of the residues, which serves as the threshold for a valid model. ERRAT was used to analyze the non-bonded atomic interactions, yielding an overall quality score of 95%, signifying appropriate model quality. Furthermore, WHAT IF was used to identify structural deviations such as unconventional bond angles, bond lengths, atomic collisions, and inaccurate crystal structure. MolProbity determined all-atom interactions, hydrogen bond networks, and steric incompatibilities.

### 4.5. Binding Site Prediction

GnRH-I receptor binding pockets were predicted using a machine learning-based technique, PrankWeb (https://prankweb.cz/, accessed on 20 August 2024). The binding site indicates the distribution of amino acid residues in the active pocket, which are catalytic residues.

### 4.6. Ligand Preparation and Docking

Studies using in silico docking were carried out using the GnRH-I receptor sequence, which was entirely isolated from the *T. ilisha*. Modeling was conducted with the modeler package 10.5. We acquired a comprehensive GnRH-I receptor sequence derived using AlphaFold3 in our study [42].

The Swiss ADME web server was employed to assess the drug-like properties of the specified chemical compounds. The examined compounds were analyzed for their characteristics, including molecular weight (MW), hydrogen bond acceptors, topological polar surface area (TPSA), donors, the number of rotatable bonds, and the water partition coefficient (WLOGP). The pharmacokinetic properties of phytocompounds encompass a toxicity screening, as it is imperative to ascertain whether a compound poses a risk when utilized as a therapeutic agent.

A total of 100 compounds were retrieved from the PubChem database and used collectively as ligands in this study; after ADME screening, 16 compounds were selected for further analysis. Molecular docking techniques were used using Auto Dock Vina 4.2 to determine the docking effectiveness of ligands with the target protein. The initial library of 100 ligands was curated from PubChem using keyword searches (‘GnRH analog,’ ‘LHRH peptide,’ ‘GnRH inhibitor’) and filtered by drug-likeness (MW < 500 Da, ≤10 rotatable bonds), documented receptor relevance, and structural diversity. Among the ligands, Leuprolide, a well-known GnRH analog with clinically validated binding properties, was included as a reference compound to serve as a control for docking affinity comparison. Following the removal of false positives and the conformational change, the redocking that received another score was subjected to enhanced docking procedures. [43,44]. Redocking was performed for validation by comparing predicted and known poses of reference ligands (e.g., Leuprolide). RMSD values < 2 Å indicated successful pose reproduction. Docking was carried out with AutoDock Vina using a grid box of 25.82 × 48.11 × 17.75 Å centered on the predicted binding pocket. Each ligand was docked with 10 poses generated per run.

### 4.7. Molecular Dynamics Simulation (MD)

Desmond (D. E. Shaw Research: Resources) was utilized to perform molecular dynamics simulations of the five leading protein–ligand complexes. The Protein Preparation Wizard was employed to prepare the aforementioned complexes. The amino acid residues were maintained in their predominant state at pH 7.0, while the GnRH receptor was encapsulated with acetyl and methylamide. Initially, the complex structure underwent minimization through the steepest descent energy minimization technique, followed by controlled heating to 310 K. To minimize the presence of heavy atoms in the solution, the Berendsen NVT ensemble was implemented to simulate the system, with the temperature regulated at 310 K. The molecular dynamics simulation proceeded at a temperature of 300 K, under a pressure of 1 atm, and with a thermostat relaxation time of 200 ps (NPT) within an isothermal and isobaric ensemble. The MD simulation was executed for a duration of 100 ns at a temperature of 300 K and a pressure of 1 atm, utilizing a Nose-Hoover thermostat and a semi-isotropic Parrinello-Rahman barostat, respectively. Comprehensive imaging from the MD simulation was conducted at 50 ps intervals. Post-MD analysis encompassed 2000 snapshots of the MD trajectories, focusing on flexibility, dynamical stability, and studies of intermolecular interactions.

### 4.8. Phylogenetic Analysis

The obtained sequences were aligned with homologous sequences from other teleost fishes using iTOL v7 web server. Phylogenetic trees were constructed using neighbor-joining or maximum likelihood methods to infer the evolutionary relationships among the GnRH isoforms.

## 5. Conclusions

The study elucidates the reproductive neuroendocrinology of *T. ilisha*, a clupeiform fish species, through the identification and characterization of the Gonadotropin-Releasing Hormone (GnRH-I) isoforms within the brain. The findings underscore the intricacy of neuroendocrine regulation in clupeiforms, revealing the complex interplay of various GnRH isoforms in regulating reproductive behaviors and physiological processes. The presence of multiple GnRH isoforms in *T. ilisha* indicates a sophisticated mechanism for finely tuning reproductive responses to environmental stimuli and social interactions. Furthermore, the study holds practical implications for the conservation and management of clupeiform fish populations, ensuring the sustainability of ecologically and economically significant species.

Two notably effective methods for examining the GnRH gene in *T. ilisha* include protein modeling and molecular docking. These methodologies facilitate precise predictions of ligand–protein interactions, thereby aiding in the identification of potential ligands for further research and development. Such computational techniques have proven to be invaluable in the study of fish reproduction and offer promising prospects for the formulation of treatments addressing disorders related to fish reproductive health. However, to validate these predictions and achieve a comprehensive understanding of the structure and function of the GnRH gene in *T. ilisha*, it is imperative to integrate computational outcomes with experimental data. Ongoing research in this domain is anticipated to yield advancements in the field of fish reproduction studies and generate opportunities for innovative strategies in aquaculture and fisheries management.

## Figures and Tables

**Figure 1 ijms-26-06098-f001:**
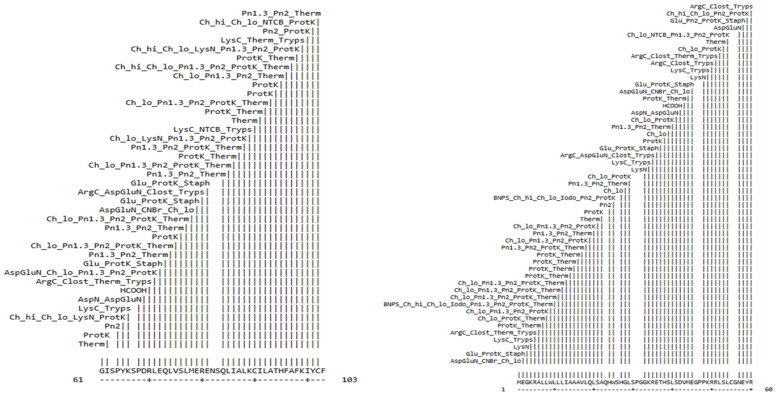
Cleavage site of the protein sequence’s chemicals and enzymes.

**Figure 2 ijms-26-06098-f002:**
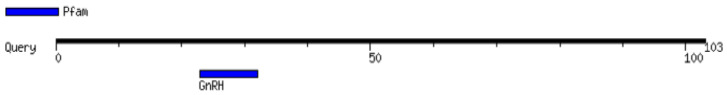
Motif region of GnRH protein predicted by motif tool.

**Figure 3 ijms-26-06098-f003:**
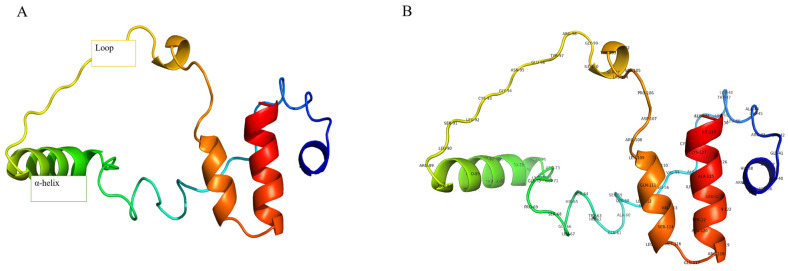
GnRH protein 3D structure (**A**), protein 3D structure with residues (**B**).

**Figure 4 ijms-26-06098-f004:**
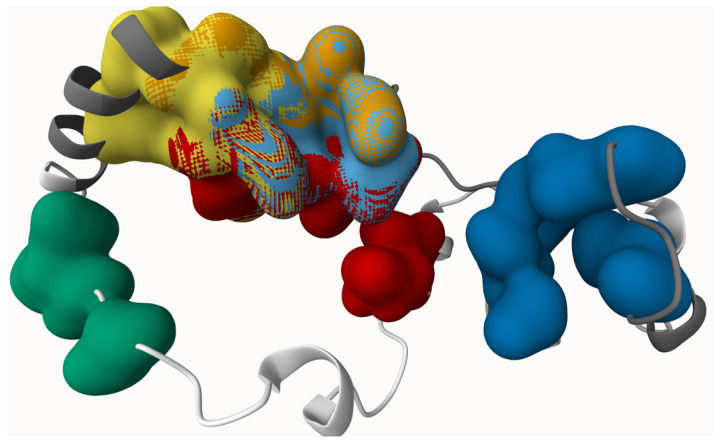
Binding pocket colors are as follows: Red (pocket 1), Green (pocket 2), Mustard (pocket 3), Copper sulfate (pocket 4), Blue (pocket 5), and Yellow (pocket 6).

**Figure 5 ijms-26-06098-f005:**
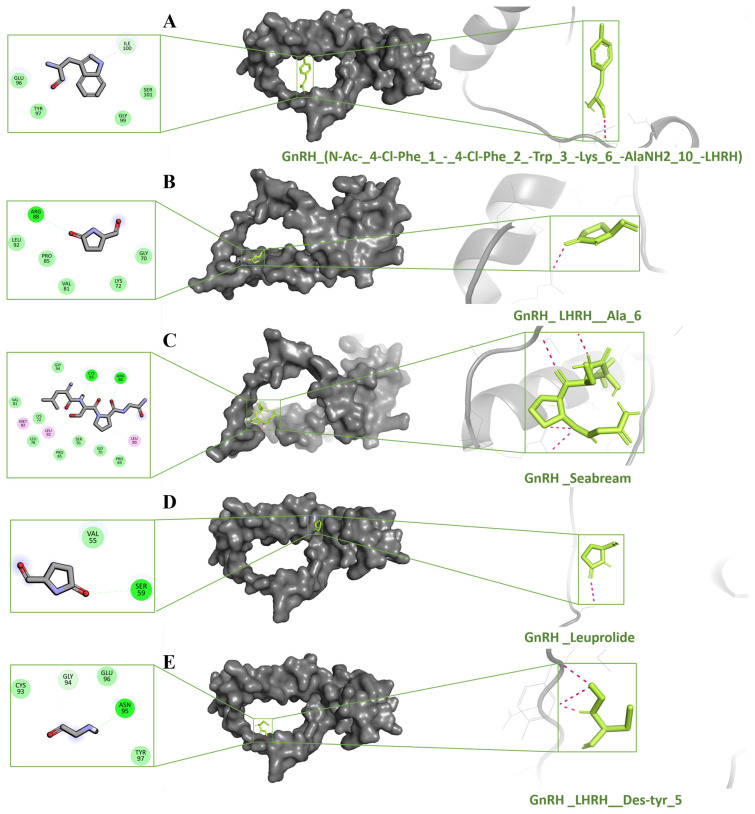
The top five ligands (**A**–**E**) with GnRH protein reveal the interactions. To produce a structurally overlaid image of the top-ranked GnRH–ligand complexes, hydrogen bonds are shown as dotted lines (hot pink).

**Figure 6 ijms-26-06098-f006:**
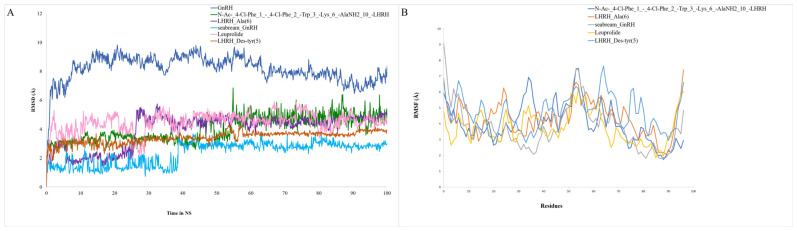
(**A**) The backbone RMSD of the five systems, including the complex of GnRH, was analyzed during 500 ns MD simulations. (**B**) During the MD simulation, the Cα-root mean squared fluctuation (RMSF) profile of each amino acid of GnRH was analyzed.

**Figure 7 ijms-26-06098-f007:**
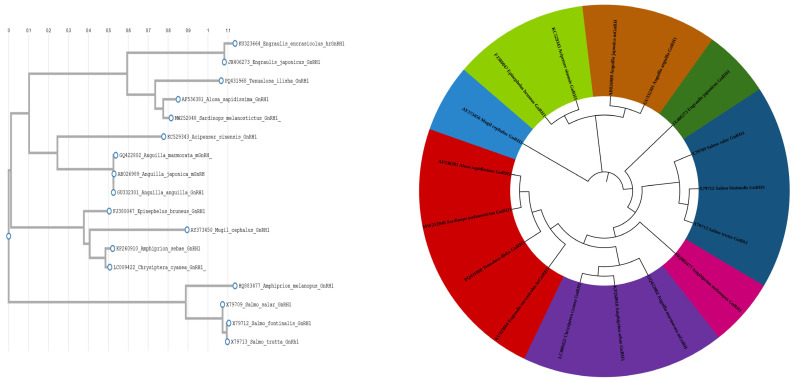
The evolutionary homology of the *T. ilisha* GnRH1 nucleotide sequence. Branches were validated by cluster in the taxa in 1000 replicates of bootstrap, represented as a percentage shown in each branch node. The phylogenetic tree was analyzed using the neighbor-joining method for branching.

**Figure 8 ijms-26-06098-f008:**
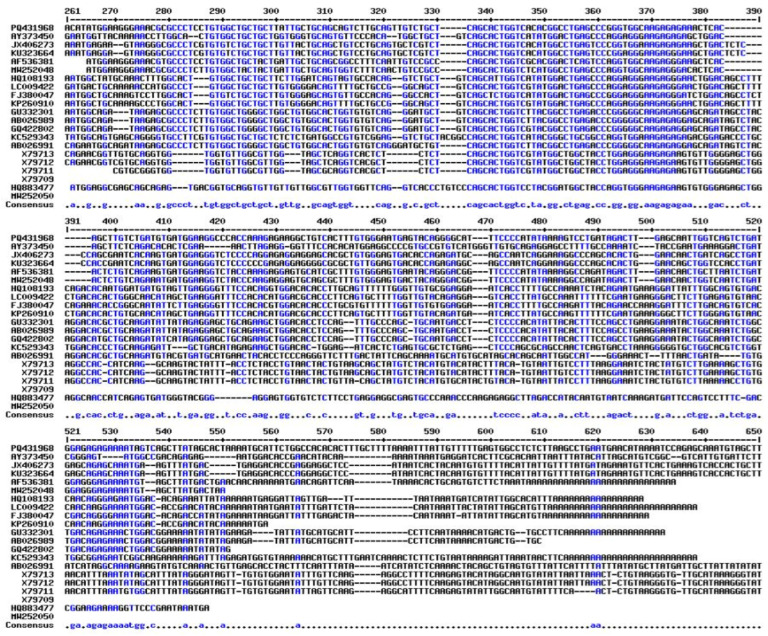
Multiple sequence alignment (MSA) of the GnRH peptide (or a related gene/protein region) in *Tenualosa ilisha* compared with homologous sequences from other related clupeiform species.

**Figure 9 ijms-26-06098-f009:**
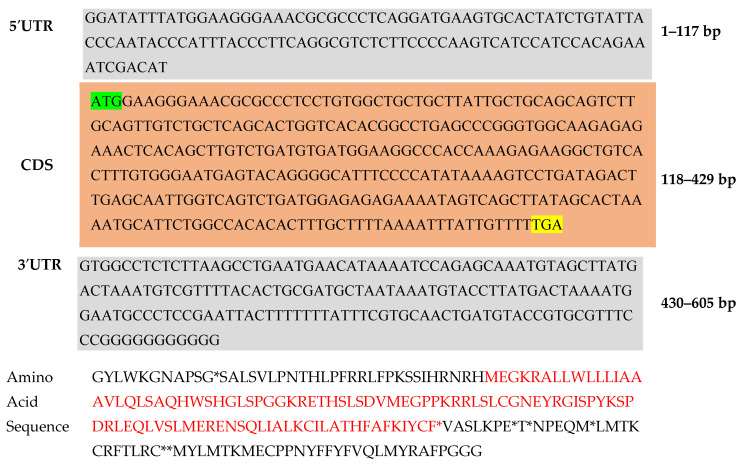
*Tenualosa ilisha* Gonadotropin-Releasing Hormone (GnRH-I) mRNA complete CDS 605 bp nucleotide collection (up), 103 protein series (down) of *T. ilisha*, green color showing for start codon and yellow color for stop codon. (the asterisk symbol * indicates a stop codon—a signal to end translation during protein synthesis.)

**Table 1 ijms-26-06098-t001:** In-detail cleavage site of the enzymes and chemicals in the GnRH protein sequence.

Position of Cleavage Site	Name of Cleaving Enzyme	Resulting Peptide Sequence	Peptide Length (aa)	Peptide Mass (Da)
1	Asp-N endopeptidase + N-terminal Glu CNBr Chymotrypsin-low specificity (C-term to [FYWML], not before P)	M	1	149.208
2	Glutamyl endopeptidase Proteinase K Staphylococcal peptidase I	E	1	147.131
3	LysN	G	1	75.067
4	LysC Trypsin	K	1	146.189
5	Arg-C proteinase Clostripain Thermolysin Trypsin	R	1	174.203
6	Proteinase K Thermolysin	A	1	89.094
7	Chymotrypsin-low specificity (C-term to [FYWML], not before P) Proteinase K Thermolysin	L	1	131.175
8	Chymotrypsin-low specificity (C-term to [FYWML], not before P) Pepsin (pH 1.3) Pepsin (pH > 2) Proteinase K	L	1	131.175
9	BNPS-Skatole Chymotrypsin-high specificity (C-term to [FYW], not before P) Chymotrypsin-low specificity (C-term to [FYWML], not before P) Iodosobenzoic acid Pepsin (pH 1.3) Pepsin (pH > 2) Proteinase K Thermolysin	W	1	204.228
10	Chymotrypsin-low specificity (C-term to [FYWML], not before P) Pepsin (pH 1.3) Pepsin (pH > 2) Proteinase K Thermolysin	L	1	131.175
11	Chymotrypsin-low specificity (C-term to [FYWML], not before P) Pepsin (pH 1.3) Pepsin (pH > 2) Proteinase K Thermolysin	L	1	131.175
12	Chymotrypsin-low specificity (C-term to [FYWML], not before P) Pepsin (pH 1.3) Pepsin (pH > 2) Proteinase K Thermolysin	L	1	137.175
13	Proteinase K Thermolysin	I	1	131.175
14	Proteinase K Thermolysin	A	1	89.094
15	Proteinase K Thermolysin	A	1	89.094
16	Proteinase K Thermolysin	A	1	89.094
17	Pepsin (pH 1.3) Pepsin (pH > 2) Proteinase K Thermolysin	V	1	117.148
18	Chymotrypsin-low specificity (C-term to [FYWML], not before P) Pepsin (pH 1.3) Pepsin (pH > 2) Proteinase K	L	1	131.175
19	Pepsin (pH 1.3) Pepsin (pH > 2) Thermolysin	Q	1	146.146
20	Chymotrypsin-low specificity (C-term to [FYWML], not before P) Pepsin (pH 1.3) Pepsin (pH > 2) Proteinase K	L	1	131.175
21	Thermolysin	S	1	105.093
22	Proteinase K	A	1	89.094
24	Pepsin (pH > 2)	Q H	2	283.287
25	BNPS-Skatole Chymotrypsin-high specificity (C-term to [FYW], not before P) Chymotrypsin-low specificity (C-term to [FYWML], not before P) Iodosobenzoic acid Pepsin (pH > 2) Proteinase K	W	1	204.228
27	Chymotrypsin-low specificity (C-term to [FYWML], not before P)	SH	2	242.235
28	Pepsin (pH 1.3) Pepsin (pH > 2) Thermolysin	G	1	75.067
29	Chymotrypsin-low specificity (C-term to [FYWML], not before P) Proteinase K	L	1	131.175
33	LysN	SPGG	4	316.314
34	LysC Trypsin	K	1	146.189
35	Arg-C proteinase Asp-N endopeptidase + N-terminal Glu Clostripain Trypsin	R	1	174.203
36	Glutamyl endopeptidase Proteinase K Staphylococcal peptidase I	E	1	147.131
37	Proteinase K	T	1	119.120
38	Chymotrypsin-low specificity (C-term to [FYWML], not before P)	H	1	155.156
39	Pepsin (pH 1.3) Pepsin (pH > 2) Thermolysin	S	1	105.093
40	Chymotrypsin-low specificity (C-term to [FYWML], not before P) Proteinase K	L	1	131.175
41	Asp-N endopeptidase Asp-N endopeptidase + N-terminal Glu	S	1	105.093
42	Formic acid	D	1	133.104
43	Proteinase K Thermolysin	V	1	117.148
44	Asp-N endopeptidase + N-terminal Glu CNBr Chymotrypsin-low specificity (C-term to [FYWML], not before P)	M	1	149.208
45	Glutamyl endopeptidase Proteinase K Staphylococcal peptidase I	E	1	147.131
48	LysN	GPP	3	269.301
49	LysC Trypsin	K	1	146.189
50	Arg-C proteinase Clostripain Trypsin	R	1	154.203
51	Arg-C proteinase Clostripain Thermolysin Trypsin	R	1	174.203
52	Chymotrypsin-low specificity (C-term to [FYWML], not before P) Proteinase K	L	1	131.175
53	Thermolysin	S	1	105.093
54	Chymotrypsin-low specificity (C-term to [FYWML], not before P) NTCB (2-nitro-5-thiocyanobenzoic acid) Pepsin (pH 1.3) Pepsin (pH > 2) Proteinase K	L	1	131.175
57	Asp-N endopeptidase + N-terminal Glu	CGN	3	192.310
58	Glutamyl endopeptidase Pepsin (pH > 2) Proteinase K Staphylococcal peptidase I	E	1	147.131
59	Chymotrypsin-high specificity (C-term to [FYW], not before P) Chymotrypsin-low specificity (C-term to [FYWML], not before P) Pepsin (pH > 2) Proteinase K	Y	1	181.191
60	Arg-C proteinase Clostripain Trypsin	R	1	174.203
61	Thermolysin	G	1	75.067
62	Proteinase K	I	1	171.175
64	Pepsin (pH > 2)	SP	2	202.210
65	Chymotrypsin-high specificity (C-term to [FYW], not before P) Chymotrypsin-low specificity (C-term to [FYWML], not before P) LysN Proteinase K	Y	1	181.191
66	LysC Trypsin	K	1	146.189
68	Asp-N endopeptidase Asp-N endopeptidase + N-terminal Glu	SP	2	202.210
69	Formic acid	D	1	133.104
70	Arg-C proteinase Clostripain Thermolysin Trypsin	R	1	174.203
71	Asp-N endopeptidase + N-terminal Glu Chymotrypsin-low specificity (C-term to [FYWML], not before P) Pepsin (pH 1.3) Pepsin (pH > 2) Proteinase K	L	1	131.138
72	Glutamyl endopeptidase Proteinase K Staphylococcal peptidase I	E	1	147.131
73	Pepsin (pH 1.3) Pepsin (pH > 2) Thermolysin	Q	1	146.146
74	Chymotrypsin-low specificity (C-term to [FYWML], not before P) Pepsin (pH 1.3) Pepsin (pH > 2) Proteinase K Thermolysin	L	1	131.175
75	Proteinase K	V	1	117.148
76	Pepsin (pH 1.3) Pepsin (pH > 2) Thermolysin	S	1	105.093
77	Chymotrypsin-low specificity (C-term to [FYWML], not before P) Pepsin (pH 1.3) Pepsin (pH > 2) Proteinase K Thermolysin	L	1	131.175
78	Asp-N endopeptidase + N-terminal Glu CNBr Chymotrypsin-low specificity (C-term to [FYWML], not before P)	M	1	149.208
79	Glutamyl endopeptidase Proteinase K Staphylococcal peptidase I	E	1	149.131
80	Arg-C proteinase Asp-N endopeptidase + N-terminal Glu Clostripain Trypsin	R	1	174.203
81	Glutamyl endopeptidase Proteinase K Staphylococcal peptidase I	E	1	147.131
84	Pepsin (pH 1.3) Pepsin (pH > 2) Thermolysin	NSQ	3	347.328
85	Chymotrypsin-low specificity (C-term to [FYWML], not before P) Pepsin (pH 1.3) Pepsin (pH > 2) Proteinase K Thermolysin	L	1	131.175
86	Proteinase K Thermolysin	I	1	131.175
87	Pepsin (pH 1.3) Pepsin (pH > 2) Proteinase K Thermolysin	A	1	89.094
88	Chymotrypsin-low specificity (C-term to [FYWML], not before P) LysN Pepsin (pH 1.3) Pepsin (pH > 2) Proteinase K	L	1	131.0175
89	LysC NTCB (2-nitro-5-thiocyanobenzoic acid) Trypsin	K	1	146.189
90	Thermolysin	C	1	121.154
91	Proteinase K Thermolysin	I	1	131.175
92	Chymotrypsin-low specificity (C-term to [FYWML], not before P) Pepsin (pH 1.3) Pepsin (pH > 2) Proteinase K Thermolysin	L	1	131.175
93	Proteinase K	A	1	89.094
94	Proteinase K	T	1	191.120
95	Chymotrypsin-low specificity (C-term to [FYWML], not before P) Pepsin (pH 1.3) Pepsin (pH > 2) Thermolysin	H	1	155.156
96	Chymotrypsin-high specificity (C-term to [FYW], not before P) Chymotrypsin-low specificity (C-term to [FYWML], not before P) Pepsin (pH 1.3) Pepsin (pH > 2) Proteinase K Thermolysin	F	1	165.189
97	Proteinase K Thermolysin	A	1	89.094
98	Chymotrypsin-high specificity (C-term to [FYW], not before P) Chymotrypsin-low specificity (C-term to [FYWML], not before P) LysN Pepsin (pH 1.3) Pepsin (pH > 2) Proteinase K	F	1	165.192
99	LysC Thermolysin Trypsin	K	1	146.189
100	Pepsin (pH > 2) Proteinase K	I	1	131.175
101	Chymotrypsin-high specificity (C-term to [FYW], not before P) Chymotrypsin-low specificity (C-term to [FYWML], not before P) NTCB (2-nitro-5-thiocyanobenzoic acid) Proteinase K	Y	1	181.191
102	Pepsin (pH 1.3) Pepsin (pH > 2) Thermolysin	C	1	121.154
103	**end of sequence**	F	1	165.192

**Table 2 ijms-26-06098-t002:** *Tenualosa ilisha* GnRH decapeptide sequence comparison with other clupeiformes fishes. GnRH peptides are similar to those of other clupeiformes fishes.

Sl.No.	Species Name	Family	Gene Name	Accessions Name	Peptides
1	*Tenualosa ilisha*	Clupeidae	*GnRH1*	PQ431968	QHWS**H**GL**S**PG
2	*Alosa sapidissima*	Clupeidae	*GnRH1*	AF536381	QHWS**H**GL**S**PG
3	*Sardinops melanostictus*	Clupeidae	*GnRH1*	MW252048	QHWS**H**GL**S**PG
4	*Engraulis japonicus*	Engraulidae	*GnRH1*	JX406273	QHWS**H**GL**S**PG
5	*Engraulis encrasicolus*	Engraulidae	*GnRH1*	KU323664	QHWS**H**GL**S**PG
6	*Anguilla anguilla*	Anguillidae	*GnRH1*	GU332301	QHWS**Y**GL**R**PG
7	*Anguilla marmorata*	Anguillidae	*mGnRH*	GQ422802	QHWS**Y**GL**R**PG
8	*Anguilla japonica*	Anguillidae	*mGnRH*	AB026989	QHWS**Y**GL**R**PG
9	*Acipenser sinensis*	Acipenseridae	*GnRH1*	KC529343	QHWS**Y**GL**R**PG
10	*Mugil cephalus*	Mugilidae	*GnRH1*	AY373450	QHWS**Y**GL**S**PG
11	*Chrysiptera cyanea*	Pomacentridae	*GnRH1*	LC009422	QHWS**Y**GL**S**PG
12	*Amphiprion sebae*	Pomacentridae	*GnRH1*	KP260910	QHWS**Y**GL**S**PG
13	*Amphiprion melanopus*	Pomacentridae	*GnRH1*	HQ883477	QHWS**Y**GW**L**PG
14	*Salmo trutta*	Salmonidae	*GnRH1*	X79713	QHWS**Y**GW**L**PG
15	*Salmo salar*	Salmonidae	*GnRH1*	X79709	QHWS**Y**GW**L**PG
16	*Salmo fontinalis*	Salmonidae	*GnRH1*	X79712	QHWSYGW**L**PG

Note: *Tenualosa ilisha* peptide similar with all clupeidae family but most differences are in positions 6–9 of the peptide, which may affect receptor interaction or regulatory function.

**Table 3 ijms-26-06098-t003:** The PrankWeb finding for GnRH includes potential binding sites and anticipated amino acids.

Pocket	Amino Acids Make up the Pocket	Grid Center		
		X	Y	Z
1	LEU109, GLN111, LEU112, LEU58, TRP63, SER64, SER68, PRO69, GLY70, GLY71, LYS72, ARG73	25.7012	48.0389	17.753
2	HIS65, GLY66, SER68, PRO69, LYS72, ARG73, GLU74, VAL81	23.5412	46.6843	7.1108
3	SER59, HIS62, HIS65, GLY66, GLU74,	29.1709	52.7545	5.1961
4	LEU58, SER59, HIS62, SER64, HIS65, ARG73	28.3709	54.1002	12.4156
5	SER91, LEU92, CYS93, ASN95	7.4175	45.3487	24.4984
6	LEU123, LYS127, ALA131, HIS38, LEU46, ALA52, ALA53, ALA54	43.5678	60.8414	24.49

**Table 4 ijms-26-06098-t004:** Ligands based on their ADME properties, bioactive scores, and toxicity parameters.

Sl.No.	Compounds	MW (<500)	TPSA (≤140)	nOHNH (≤5)	nON (≤5)	WLOGP (≤5.88)	Nrotb (≤10)
1	Seabream	413.18	130.46	5	5	−5.21	7
2	LHRH, Lys(6)-	453.41	100.65	7	6	−4.65	4
3	LHRH, leu(6)-leu(N-alpha-Me)(7)-N-Et-pronh2(9)-	423.42	122.75	4	4	−1.86	4
4	LHRH, his(6)-N-Et-pronh2(9)-	433.38	160.22	6	5	−3.28	4
5	LHRH, his(5)-trp(7)-tyr(8)-	436.3	154.7	6	5	−3.56	8
6	LHRH, gln(1)-des-his(2)-phe(6)-N-Et-pronh2(9)-	457.44	126.08	4	5	0.06	4
7	LHRH, Des-tyr(5)-	319.12	125.3	4	3	−5.58	5
8	LHRH (2–10), Trp(6)-	400.35	158.24	6	4	−2.11	9
9	Leuprolide	409.4	131.54	5	4	−2.2	4
10	H-Pyr-His-Trp-Ser-Tyr-Gly-OH	259.77	196.83	1	1	−2.35	3
11	Gppt-LHRH	418.6	179.55	7	5	−1.96	5
12	GnRH antagonist 2	207.57	150.43	2	1	2.71	1
13	Ac-Abu-Phe(4-F)-Trp-Asp(1)-Gln-D-Arg-Leu-D-Lys-Pro-N(1)Gly-OH	473.41	177.09	4	7	−4.14	3
14	Ac-Abu-Phe(4-Cl)-Trp-Asp-D-Cys(1)-D-Arg-D-Leu-D-Cys(1)-Pro-Ala-NH2	468.89	193.39	4	4	0.35	3
15	N-Ac-(4-Cl-Phe)(1)-(4-Cl-Phe)(2)-Trp(3)-Lys(6)-AlaNH2(10)-LHRH	371.19	142.45	5	4	−2.45	7
16	3-ylpropanoyl amino-4-(2R,5S,8S,20R)-8-(2R)-2-(2S)-1-amino-1-oxopropan-2-ylcarbamoylpyrr	436.88	115.16	5	8	−2.6	3

**Table 5 ijms-26-06098-t005:** Binding affinity of ligands to the GnRH protein and their shape.

Sl.No.	Ligand	CID	Binding Affinity
1	N-Ac-(4-Cl-Phe)(1)-(4-Cl-Phe)(2)-Trp(3)-Lys(6)-AlaNH2(10)-LHRH	25078179	−7.5
2	LHRH_LYS(6)	162695	−6.7
3	Seabream_GnRH	118856782	−5.9
4	Leuprolide	657181	−5.8
5	LHRH_Des-tyr(5)	3081503	−5.6
6	LHRH, leu(6)-leu(N-alpha-Me)(7)-N-Et-pronh2(9)	173522	−5.5
7	H-Pyr-His-Trp-Ser-Tyr-Gly-OH	13054089	−5.5
8	Gppt-LHRH	16132140	−5.4
9	LHRH, his(6)-N-Et-pronh2(9)	164329	−5.4
10	GnRH antagonist 2	91755013	−5.3
11	LHRH, his(5)-trp(7)-tyr(8)-	16130955	−5.3
12	LHRH, gln(1)-des-his(2)-phe(6)-N-Et-pronh2(9)	25079029	−5.2
13	LHRH (2–10), Trp(6)	25079046	−5.2
14	Ac-Abu-Phe(4-Cl)-Trp-Asp-D-Cys(1)-D-Arg-D-Leu-D-Cys(1)-Pro-Ala-NH2	44378909	−5
15	Ac-Abu-Phe(4-F)-Trp-Asp(1)-Gln-D-Arg-Leu-D-Lys-Pro-N(1)Gly-OH	44378908	−4.9
16	(3S)-3-[[(2S)-2-[[(2S)-2-[[(2S)-2-acetamidobutanoyl]amino]-3-(4-chlorophenyl)propanoyl]amino]-3-pyridin-3-ylpropanoyl]amino]-4-[[(2R,5S,8S,20R)-8-[(2R)-2-[[(2S)-1-amino-1-oxopropan-2-yl]carbamoyl]pyrr	44378966	−4.7

## Data Availability

The original contributions presented in the study are publicly available. This data can be found here: NCBI GenBank under the accession number PQ431968 and are available from the corresponding author on reasonable request.

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
