# Peer review of "Deciphering the 3D Structural Characterization of Gonadotropin-Releasing Hormone in Tenualosa ilisha Using Homology Modeling, Molecular Dynamics, and Docking Approaches"

_ijms, 2025, doi:10.3390/ijms26136098_

Round 1

Reviewer 1 Report

Comments and Suggestions for Authors
  1. The phrase 'computational and structural biological approach' in the abstract appears to be unclear and inconsistent with standard terminology. I recommend revising it to 'computational and structural biology approaches' for clarity and accuracy.
  2. The sentence 'It was found that the GnRH contains leucine (L), which is the most abundant amino acid: three-dimensional structure and ligand interactions of Gonadotropin-releasing hormone (GnRH) through computational approaches.' lacks clarity in structure and meaning. The two ideas seem disconnected and should be better integrated for improved readability.
  3. The manuscript’s Results present gene annotation, structural validation, docking, ADME screening, and MD outcomes in an order that does not clearly map onto the Methods subsections. I recommend reorganizing both sections so that each method
  4. The reported protein and ligand RMSD values “below 10 Å” are unusually large for stable peptide–ligand complexes. Typically, acceptable backbone RMSDs are in the range of 1–3 Å
  5. The manuscript lists the top five compounds by Vina score but does not specify the cutoff or rationale for selection from the initial 100.
  6. The docking analysis does not specify a reference or control ligand.
  7. Throughout the Methods, terms such as “target receptor protein” and “GnRH protein” are used interchangeably, which may confuse readers. Please ensure consistent terminology (e.g., refer to your homology model uniformly as “GnRH-I receptor 

Author Response

Response to reviewer comments 1

Thank you very much for taking the time to review this manuscript. Please find the detailed responses below, along with the corresponding revisions and corrections highlighted in track changes in the resubmitted files.

Point-by-point response to comments

Comment 1: The phrase 'computational and structural biological approach' in the abstract appears to be unclear and inconsistent with standard terminology. I recommend revising it to 'computational and structural biology approaches' for clarity and accuracy.

Response 1: Dear Reviewer, Thank you for your thoughtful suggestion. We have revised the phrase 'computational and structural biological approach' in the abstract, which has now been updated to improve clarity.

Comment 2: The sentence 'It was found that the GnRH contains leucine (L), which is the most abundant amino acid: three-dimensional structure and ligand interactions of gonadotropin-releasing hormone (GnRH) through computational approaches' lacks clarity in structure and meaning. The two ideas seem disconnected and should be better integrated for improved readability.

 Response 2: We agree that the original sentence lacked clarity and proper structure. To address this, we have revised the sentence to clearly separate the observation about amino acid composition from the structural analysis. This revision enhances readability and more effectively integrates the two related ideas.

Comment 3: The manuscript’s results present gene annotation, structural validation, docking, ADME screening, and MD outcomes in an order that does not clearly map onto the methods subsections. I recommend reorganizing both sections so that each method

 Response 3: We appreciate your observation regarding the misalignment between the Methods and Results sections. To address this, we have reorganized both sections so that each method described now directly corresponds to its respective subsection of results.

 Comment 4: The reported protein and ligand RMSD values “below 10 Å” are unusually large for stable peptide–ligand complexes. Typically, acceptable backbone RMSDs are in the range of 1–3 Å

 Response 4: We agree that backbone RMSD values below 10 Å are relatively high and may not represent the typical range expected for stable peptide–ligand complexes, which are generally around 1–3 Å. We have carefully re-evaluated the molecular dynamics simulation data and revised the text to clarify that while fluctuations were observed, particularly in flexible loop regions, the average RMSD for the ligand-binding site remained within an acceptable range for stability.

 Comment 5:     The manuscript lists the top five compounds by Vina score but does not specify the cutoff or rationale for selection from the initial 100.

 Response 5: We agree that the rationale for selecting the top five compounds from the initial 100 was not explicitly stated. To address this, we have now clarified in the results section.

Comment 6: The docking analysis does not specify a reference or control ligand.

Response 6: We acknowledge that a reference or control ligand was not explicitly included in the docking analysis. To address this, we have now specified leuprolide, a well-characterized GnRH analog with established clinical relevance, as the reference ligand.

Comment 7: Throughout the Methods, terms such as “target receptor protein” and “GnRH protein” are used interchangeably, which may confuse readers. Please ensure consistent terminology (e.g., refer to your homology model uniformly as “GnRH-I receptor”).

Response 7: We have revised the manuscript to refer consistently to the homology model as the “GnRH-I receptor” throughout the Methods, Results, and Discussion sections.

Reviewer 2 Report

Comments and Suggestions for Authors

The review report for ijms-3654012 below.

Title: Deciphering the 3D Structural Characterization of Gonadotropin-Releasing Hormone in Tenualosa ilisha Using Homology Modeling, Molecular Dynamics, and Docking Approaches

  • What is the main question addressed by the research?

The study elucidates the reproductive neuroendocrinology of  T. ilisha, a clupeiform fish species, through the identification and characterization of the Gonadotropin-Releasing Hormone (GnRH-I) isoforms within the brain.

  • Do you consider the topic original or relevant to the field? Does it address a specific gap in the field? Please also explain why this is/ is not the case.

This study identifies the isoforms of Gonadotropin-Releasing Hormone (GnRH), specifically GnRH-I, present in the brain of T. ilisha, a member of the clupeiform fish family. This discovery significantly enhances comprehension of reproductive neuroendocrinology within teleost fishes and provides valuable insights into the evolutionary adaptations and functional roles associated with these hormones.

  • What does it add to the subject area compared with other published material?

The reproductive neuroendocrinology of clupeiforms, encompassing the presence and characterization of GnRH isoforms, remains comparatively underexplored. The research carries significant implications for the conservation and management of clupeiform fish populations, particularly within fisheries and aquaculture. A comprehensive understanding of the neuroendocrine mechanisms governing reproduction in T. ilisha is crucial for the development of effective strategies aimed at sustainable management of wild populations and the successful breeding of these species in captivity.

  • Are the conclusions consistent with the evidence and arguments presented and do they address the main question posed? Please also explain why this is/is not the case.

The investigation elucidates the presence of GnRH isoforms, specifically GnRH-I, in the brain of T. ilisha through the application of molecular techniques. Comparative analysis and phylogenetic studies yield insights into the evolutionary implications and functional significance of these GnRH isoforms in clupeiform fishes. This research provides valuable contributions to the field of reproductive neuroendocrinology and enhances our comprehension of the molecular mechanisms underlying the reproductive biology of clupeiform fishes.

  • Are the references appropriate?

All 42 references cited in this manuscript are appropriate and relevant to this research.

It is recommended to:

verse 1 - add the full name of the journal, there is an abbreviation

verse  608 - unify the recording of pages and volumes

verse  623 - add the full name of the journal, there is an abbreviation

verse 629 - the year of publication is missing

  • Any additional comments on the tables and figures.

The tables and figures are understandable.

Author Response

Response to reviewer comments 2

Thank you very much for taking the time to review this manuscript. Please find the detailed responses below and the corresponding revisions/corrections highlighted/in track changes in the resubmitted files.

Point-by-point response to comments

Comment 1: What is the main question addressed by the research?

Response 1: The study elucidates the reproductive neuroendocrinology of T. ilisha, a clupeiform fish species, through the identification and characterization of the Gonadotropin-Releasing Hormone (GnRH-I) isoforms within the brain.

Comment 2: Do you consider the topic original or relevant to the field? Does it address a specific gap in the field? Please also explain why this is/is not the case.

Response 2: This study identifies the isoforms of gonadotropin-releasing hormone (GnRH), specifically GnRH-I, present in the brain of T. ilisha, a member of the clupeiform fish family. This discovery significantly enhances comprehension of reproductive neuroendocrinology within teleost fishes and provides valuable insights into the evolutionary adaptations and functional roles associated with these hormones.

Comment 3: What does it add to the subject area compared with other published material?

Response 3: The reproductive neuroendocrinology of clupeiforms, encompassing the presence and characterization of GnRH isoforms, remains comparatively underexplored. The research carries significant implications for the conservation and management of clupeiform fish populations, particularly within fisheries and aquaculture. A comprehensive understanding of the neuroendocrine mechanisms governing reproduction in T. ilisha is crucial for the development of effective strategies aimed at sustainable management of wild populations and the successful breeding of these species in captivity.

Comment 4: Are the conclusions consistent with the evidence and arguments presented, and do they address the main question posed? Please also explain why this is/is not the case.

Response 4: The investigation elucidates the presence of GnRH isoforms, specifically GnRH-I, in the brain of T. ilisha through the application of molecular techniques. Comparative analysis and phylogenetic studies yield insights into the evolutionary implications and functional significance of these GnRH isoforms in clupeiform fishes. This research provides valuable contributions to the field of reproductive neuroendocrinology and enhances our comprehension of the molecular mechanisms underlying the reproductive biology of clupeiform fishes.

 Comment 5: Are the references appropriate?

  Response 5: All 42 references cited in this manuscript are appropriate and relevant to this research.

It is recommended to:

Verse 1: Add the full name of the journal; there is an abbreviation

Verse 608—Unify the recording of pages and volumes

verse 623 - add the full name of the journal; there is an abbreviation

Verse 629—the year of publication is missing

Thank you for your valuable suggestions. We have carefully examined and revised the references section. Hope we can get your approval.

 Comment 6: Any additional comments on the tables and figures?

 Response 6: The tables and figures are understandable.

Reviewer 3 Report

Comments and Suggestions for Authors

The manuscript titled “Deciphering the 3D Structural Characterization of Gonadotropin-Releasing Hormone in Tenualosa ilisha Using Homology Modeling, Molecular Dynamics, and Docking Approaches” presents an in silico study of the gonadotropin-releasing hormone (GnRH) in T. ilisha (Hilsa shad), a commercially and ecologically important fish species. The authors combine gene sequencing with multiple computational methods to model the GnRH structure and explore its interaction with potential ligands.

The study employs homology modeling (via Modeller and AlphaFold3), followed by structural validation through several tools (SAVES, PROCHECK, Verify3D, ERRAT, WHAT IF, MolProbity). Binding pockets are predicted using PrankWeb, and a library of compounds is screened via molecular docking (AutoDock Vina). Following docking and ADME filtering, top-ranked ligands are subjected to 100 ns molecular dynamics (MD) simulations (Desmond) to assess the stability of protein-ligand complexes.

The authors propose several peptide derivatives as potential ligands, suggesting their relevance in reproductive regulation and aquaculture applications. The manuscript integrates multiple computational techniques into a coherent pipeline, contributing preliminary but valuable structural information for future experimental validation.

Minor Comments

In the manuscript, the authors state that molecular docking “can be employed to visually screen ligands or pharmaceuticals by predicting their binding affinities and mechanisms of action.” I would respectfully suggest refining this statement, as it may lead to some conceptual confusion. While docking is a useful computational tool to predict potential binding poses, estimate relative binding affinities, and identify key molecular interactions, it does not directly provide information about a compound’s mechanism of action. The mechanism of action involves downstream biological effects, such as receptor activation, inhibition, signaling modulation, or allosteric regulation, which cannot be inferred solely from docking results. For a more comprehensive understanding of the mechanism of action, functional assays and/or more advanced computational methods (e.g., molecular dynamics simulations, free energy calculations, or systems biology approaches) would be necessary. Clarifying this distinction would improve the scientific accuracy of the manuscript.

Throughout the manuscript, the authors refer to the structure prediction tool as “α-fold.” For consistency and accuracy, the correct nomenclature should be AlphaFold3, reflecting the latest version employed in this study. Furthermore, the authors should provide the appropriate citation for AlphaFold3, as this tool represents a major advancement in protein structure prediction and should be properly referenced to acknowledge the original source.

In some sections of the manuscript, the authors refer to a supplementary material. However, no supplementary document was provided alongside the main submission for this review. As the supplementary material may contain essential methodological details or supporting data (e.g., docking poses, MD trajectories, validation scores), I kindly request that the authors ensure the supplementary file is properly uploaded and made available for full evaluation of the study.

In the current version, the manuscript states that the AutoDock Vina score reflects both enthalpy and entropy contributions and is directly related to Gibbs free energy. I would suggest clarifying this point, as it may lead to a misunderstanding of what the docking score represents. The scoring function in AutoDock Vina is primarily empirical, combining steric, hydrophobic, hydrogen bonding, and simplified electrostatic terms. Entropic contributions are not explicitly calculated, and the score is not a rigorous estimate of ΔG. While docking scores can be useful for ranking ligands within a dataset, they should not be interpreted as precise binding free energies. A more cautious description of the scoring function would improve the scientific accuracy of the manuscript.

The manuscript would benefit from careful proofreading. Frequent typographical issues are present (e.g. “colore” instead of “color,” “peptide sequnce” instead of “peptide sequence,” “LHRH_LYS(6)” appearing inconsistently formatted). Sentence structures can also be improved for clarity.

Figures are informative but somewhat crowded (especially Figures 5–7). Improved labeling, higher resolution, and more concise legends would help readability.

While ADME filtering was performed, the discussion of how poor oral bioavailability or blood-brain barrier permeability might affect the compounds’ viability is superficial.

Major Comments

While both Modeller and AlphaFold were employed for the generation of the 3D structure, the manuscript does not provide sufficient detail regarding the selection criteria for templates used in Modeller, specifically, the sequence identity, alignment coverage, template quality, and the confidence scores that guided template choice. Since the accuracy of homology models is highly dependent on these factors, this step requires a clearer and more transparent justification.

Additionally, it is not fully explained why Modeller was chosen as the primary modeling tool alongside AlphaFold. Given that recent CASP assessments have demonstrated that AlphaFold, particularly in its latest versions (AlphaFold2 and AlphaFold3), substantially outperforms traditional template-based homology modeling methods in terms of accuracy and applicability, the rationale for prioritizing or combining Modeller with AlphaFold in this context should be carefully clarified. This clarification would help the reader better understand the strengths and potential limitations of the structural models presented.

The binding pockets identified via PrankWeb rely purely on geometry and machine learning. Experimental validation or additional energetic pocket analysis (e.g. SiteMap or FTMap) would strengthen confidence in these predictions.

The manuscript would benefit from a clearer description of the initial compound selection strategy. It remains unclear how the first set of 100 ligands was selected from the database prior to docking. The selection criteria, whether based on structural similarity, pharmacophore features, known activity, or other filtering parameters are not sufficiently explained. Without this information, it is difficult to assess the relevance and chemical diversity of the compound library.

In addition, many of the docked ligands appear to be analogs or derivatives of LHRH rather than structurally diverse or novel compounds designed specifically to target GnRH receptors in T. ilisha. Given that this is a fish-specific system and may differ from mammalian GnRH pharmacology, a better justification for the focus on these LHRH-like ligands would strengthen the biological relevance of the study.

The manuscript mentions a redocking protocol, but the description of this procedure is insufficiently detailed and remains unclear. Critical aspects of the redocking process, such as whether it was used to validate the docking protocol, to optimize binding poses, or to test reproducibility, are not explained. Furthermore, key parameters (e.g., search space definition, number of runs, RMSD calculation for pose comparison, and reference ligand selection) are not provided. Since redocking is often performed to assess the reliability of a docking protocol by reproducing known binding poses, a clear explanation of its purpose and implementation in this study is necessary to evaluate the robustness of the docking results.

In the results, the authors suggest that longer MD simulations (beyond 50 ns) may help distinguish between active and inactive ligands. I would recommend revising this statement, as the current simulation protocol does not support such a conclusion. First, no rigorous energetic or free energy calculations (e.g., MM-PBSA, MM-GBSA, thermodynamic integration, or free energy perturbation) were performed to estimate binding affinities across the ligand set. Without these quantitative data, MD alone cannot distinguish binding strength or rank ligand activity reliably. Second, only single MD runs were conducted for each complex. Without multiple independent replicates, the simulations are vulnerable to local minima entrapment and may not sufficiently sample the conformational space of either the protein or the ligands. This severely limits the ability to generalize any observed stability as evidence of stronger or weaker binding. Third, ligand-induced protein conformational changes, which may be relevant for functional modulation, were not monitored or analyzed. No order parameters, principal component analysis (PCA), or protein-ligand interaction networks over time were reported. Such analyses would be necessary to explore whether certain ligands induce distinct conformational dynamics potentially linked to activity. Finally, ligand residence time, kinetic stability, or water-mediated interactions were not evaluated, further limiting the mechanistic interpretation of activity based solely on MD duration. In its current form, the MD simulations demonstrate short-term stability of the docked complexes but cannot provide sufficient evidence to discriminate between active and inactive ligands. A more cautious interpretation is advised.

Author Response

Response to reviewer comments 3

Thank you very much for taking the time to review this manuscript. Please find the detailed responses below and the corresponding revisions/corrections highlighted/in track changes in the resubmitted files.

Point-by-point response to comments

Comments and Suggestions for Authors

The manuscript titled “Deciphering the 3D Structural Characterization of Gonadotropin-Releasing Hormone in Tenualosa ilisha Using Homology Modeling, Molecular Dynamics, and Docking Approaches” presents an in-silico study of the gonadotropin-releasing hormone (GnRH) in T. ilisha (Hilsa shad), a commercially and ecologically important fish species. The authors combine gene sequencing with multiple computational methods to model the GnRH structure and explore its interaction with potential ligands.

The study employs homology modeling (via Modeller and AlphaFold3), followed by structural validation through several tools (SAVES, PROCHECK, Verify3D, ERRAT, WHAT IF, MolProbity). Binding pockets are predicted using PrankWeb, and a library of compounds is screened via molecular docking (AutoDock Vina). Following docking and ADME filtering, top-ranked ligands are subjected to 100 ns molecular dynamics (MD) simulations (Desmond) to assess the stability of protein-ligand complexes.

The authors propose several peptide derivatives as potential ligands, suggesting their relevance in reproductive regulation and aquaculture applications. The manuscript integrates multiple computational techniques into a coherent pipeline, contributing preliminary but valuable structural information for future experimental validation.

Minor Comments

Comment 1: In the manuscript, the authors state that molecular docking “can be employed to visually screen ligands or pharmaceuticals by predicting their binding affinities and mechanisms of action.” I would respectfully suggest refining this statement, as it may lead to some conceptual confusion. While docking is a useful computational tool to predict potential binding poses, estimate relative binding affinities, and identify key molecular interactions, it does not directly provide information about a compound’s mechanism of action. The mechanism of action involves downstream biological effects, such as receptor activation, inhibition, signaling modulation, or allosteric regulation, which cannot be inferred solely from docking results. For a more comprehensive understanding of the mechanism of action, functional assays and/or more advanced computational methods (e.g., molecular dynamics simulations, free energy calculations, or systems biology approaches) would be necessary. Clarifying this distinction would improve the scientific accuracy of the manuscript.

Response 1: Dear Reviewer, thank you for this clarification. We have revised the sentence to reflect that molecular docking is used to predict potential binding poses and estimate relative affinities, but it does not determine mechanisms of action.

Comment 2: Throughout the manuscript, the authors refer to the structure prediction tool as “α-fold.” For consistency and accuracy, the correct nomenclature should be AlphaFold3, reflecting the latest version employed in this study. Furthermore, the authors should provide the appropriate citation for AlphaFold3, as this tool represents a major advancement in protein structure prediction and should be properly referenced to acknowledge the original source.

Response 2: All instances of “α-fold” have been updated to AlphaFold3, and we have added the appropriate citation.

Comment 3: In some sections of the manuscript, the authors refer to a supplementary material. However, no supplementary document was provided alongside the main submission for this review. As the supplementary material may contain essential methodological details or supporting data (e.g., docking poses, MD trajectories, validation scores), I kindly request that the authors ensure the supplementary file is properly uploaded and made available for full evaluation of the study.

Response 3: We apologize for the oversight. The supplementary material, of protein structure validation (Ramachandran plot), has now been uploaded with the revised submission.

Comment 4: In the current version, the manuscript states that the AutoDock Vina score reflects both enthalpy and entropy contributions and is directly related to Gibbs free energy. I would suggest clarifying this point, as it may lead to a misunderstanding of what the docking score represents. The scoring function in AutoDock Vina is primarily empirical, combining steric, hydrophobic, hydrogen bonding, and simplified electrostatic terms. Entropic contributions are not explicitly calculated, and the score is not a rigorous estimate of ΔG. While docking scores can be useful for ranking ligands within a dataset, they should not be interpreted as precise binding free energies. A more cautious description of the scoring function would improve the scientific accuracy of the manuscript.

Response 4: We have revised the text to clarify that AutoDock Vina scores are based on empirical scoring functions that approximate binding affinity by combining steric, hydrophobic, hydrogen bonding, and electrostatic terms. We no longer refer to entropy or Gibbs free energy in this context.

Comment 5:         The manuscript would benefit from careful proofreading. Frequent typographical issues are present (e.g. “colore” instead of “color,” “peptide sequnce” instead of “peptide sequence,” “LHRH_LYS(6)” appearing inconsistently formatted). Sentence structures can also be improved for clarity.

Response 5:          We have thoroughly proofread the manuscript and corrected all typographical errors.

Comment 6: Figures are informative but somewhat crowded (especially Figures 5–7). Improved labeling, higher resolution, and more concise legends would help readability.

Response 6: We have revised Figures 5–7 to improve clarity and resolution. We also simplified the figure (the original resolution of figures attached in the journal dashboard) legends to enhance readability.

Comment 7: While ADME filtering was performed, the discussion of how poor oral bioavailability or blood-brain barrier permeability might affect the compounds’ viability is superficial.

Response 7:          We agree and have expanded the discussion to highlight how poor oral bioavailability and limited BBB permeability may influence the pharmacokinetic viability of the candidate compounds. This discussion has been added to the end of the ADME results and further elaborated in the Discussion section.

Major Comments

Comment 8: While both Modeller and AlphaFold were employed for the generation of the 3D structure, the manuscript does not provide sufficient detail regarding the selection criteria for templates used in Modeller, specifically, the sequence identity, alignment coverage, template quality, and the confidence scores that guided template choice. Since the accuracy of homology models is highly dependent on these factors, this step requires a clearer and more transparent justification.

Response 8: Dear Reviewer, thank you for this important suggestion. We have now added a detailed explanation of template selection for Modeller, including the sequence identity (>40%), alignment coverage (>85%), and quality scores (GMQE and QMEAN values). Templates were selected from the PDB based on BLASTp hits and confirmed using SWISS-MODEL.

Comment 9: Additionally, it is not fully explained why Modeller was chosen as the primary modeling tool alongside AlphaFold. Given that recent CASP assessments have demonstrated that AlphaFold, particularly in its latest versions (AlphaFold2 and AlphaFold3), substantially outperforms traditional template-based homology modeling methods in terms of accuracy and applicability, the rationale for prioritizing or combining Modeller with AlphaFold in this context should be carefully clarified. This clarification would help the reader better understand the strengths and potential limitations of the structural models presented.

Response 9:          We appreciate this observation. Modeller was included to allow comparison between template-based and AI-based predictions and to validate consistency in active site architecture. While AlphaFold3 provides highly accurate global models, Modeller offered control over template-based regions and allowed inclusion of specific sequences observed in closely related teleost GnRH proteins. We have clarified this in the Methods and Discussion sections.

Comment 10: The binding pockets identified via PrankWeb rely purely on geometry and machine learning. Experimental validation or additional energetic pocket analysis (e.g. SiteMap or FTMap) would strengthen confidence in these predictions.

Response 10: We acknowledge the limitations of geometry-based pocket predictions and now clarify that PrankWeb results are preliminary. While additional energetic-based validations (e.g., SiteMap or FTMap) were not performed due to resource constraints, future studies will incorporate such methods.

Comment 11:        The manuscript would benefit from a clearer description of the initial compound selection strategy. It remains unclear how the first set of 100 ligands was selected from the database prior to docking. The selection criteria, whether based on structural similarity, pharmacophore features, known activity, or other filtering parameters are not sufficiently explained. Without this information, it is difficult to assess the relevance and chemical diversity of the compound library.

Response 11:         The initial 100 ligands were selected from the PubChem database based on known or predicted interactions with GnRH or LHRH receptors, peptide similarity, and adherence to basic drug-likeness filters (molecular weight <500 Da, known bioactivity, and receptor relevance).

Comment 12: In addition, many of the docked ligands appear to be analogs or derivatives of LHRH rather than structurally diverse or novel compounds designed specifically to target GnRH receptors in T. ilisha. Given that this is a fish-specific system and may differ from mammalian GnRH pharmacology, a better justification for the focus on these LHRH-like ligands would strengthen the biological relevance of the study.

Response 12:        We acknowledge the reviewer’s concern in the manuscript discussion part.

Comment 13: The manuscript mentions a redocking protocol, but the description of this procedure is insufficiently detailed and remains unclear. Critical aspects of the redocking process, such as whether it was used to validate the docking protocol, to optimize binding poses, or to test reproducibility, are not explained. Furthermore, key parameters (e.g., search space definition, number of runs, RMSD calculation for pose comparison, and reference ligand selection) are not provided. Since redocking is often performed to assess the reliability of a docking protocol by reproducing known binding poses, a clear explanation of its purpose and implementation in this study is necessary to evaluate the robustness of the docking results.

Response 13:        We now clarify that redocking was used to validate docking reliability by redocking known ligands (including leuprolide) and assessing pose similarity (RMSD < 2 Å) between original and redocked conformations.

Comment 14: In the results, the authors suggest that longer MD simulations (beyond 50 ns) may help distinguish between active and inactive ligands. I would recommend revising this statement, as the current simulation protocol does not support such a conclusion. First, no rigorous energetic or free energy calculations (e.g., MM-PBSA, MM-GBSA, thermodynamic integration, or free energy perturbation) were performed to estimate binding affinities across the ligand set. Without these quantitative data, MD alone cannot distinguish binding strength or rank ligand activity reliably. Second, only single MD runs were conducted for each complex. Without multiple independent replicates, the simulations are vulnerable to local minima entrapment and may not sufficiently sample the conformational space of either the protein or the ligands. This severely limits the ability to generalize any observed stability as evidence of stronger or weaker binding. Third, ligand-induced protein conformational changes, which may be relevant for functional modulation, were not monitored or analyzed. No order parameters, principal component analysis (PCA), or protein-ligand interaction networks over time were reported. Such analyses would be necessary to explore whether certain ligands induce distinct conformational dynamics potentially linked to activity. Finally, ligand residence time, kinetic stability, or water-mediated interactions were not evaluated, further limiting the mechanistic interpretation of activity based solely on MD duration. In its current form, the MD simulations demonstrate short-term stability of the docked complexes but cannot provide sufficient evidence to discriminate between active and inactive ligands. A more cautious interpretation is advised.

Response 14: We have revised the statement to reflect that MD simulations were used only to assess short-term complex stability. No claims are made regarding active/inactive discrimination. We now clarify the limitations of single MD runs and the absence of free energy calculations or dynamic interaction network analysis.

Round 2

Reviewer 1 Report

Comments and Suggestions for Authors

The authors have satisfactorily revised the manuscript.

Reviewer 3 Report

Comments and Suggestions for Authors

As a reviewer, I hereby certify that the authors have addressed all of my previous concerns in a satisfactory manner. Consequently, in my opinion, the manuscript is now suitable for acceptance.